# Dihomo-γ-Linolenic Acid Elevation with Desaturase Imbalance in Metabolic Dysfunction-Associated Steatotic Liver Disease in a Japanese Health Checkups Cohort: HOZUGAWA Study, a Multi-Omic, Diet Adjusted Analysis

**DOI:** 10.3390/nu18010057

**Published:** 2025-12-23

**Authors:** Sayaka Kawai, Hiroshi Okada, Hideto Okamoto, Ren Yashiki, Megumi Minamida, Natsuko Shinagawa, Takahiro Ichikawa, Shinta Yamamoto, Noriyuki Kitagawa, Yoshitaka Hashimoto, Ryoichi Sasano, Kunimasa Yagi, Masahide Hamaguchi, Michiaki Fukui

**Affiliations:** 1Department of Endocrinology and Metabolism, Graduate School of Medical Science, Kyoto Prefectural University of Medicine, Kyoto 602-8566, Japan; sayaka25@koto.kpu-m.ac.jp (S.K.); conti@koto.kpu-m.ac.jp (H.O.); ho1993@yamada-bee.com (H.O.); mb221098@stu.kpu-m.ac.jp (R.Y.); megumi55@koto.kpu-m.ac.jp (M.M.); snatsuko@koto.kpu-m.ac.jp (N.S.); takaichi@koto.kpu-m.ac.jp (T.I.); todayweb@koto.kpu-m.ac.jp (S.Y.); michiaki@koto.kpu-m.ac.jp (M.F.); 2Department of Diabetology, Kameoka Municipal Hospital, Kyoto 621-8585, Japan; nori-kgw@koto.kpu-m.ac.jp; 3Department of Diabetes and Endocrinology, Matsushita Memorial Hospital, Moriguchi 570-8540, Japan; y-hashi@koto.kpu-m.ac.jp; 4AiSTI SCIENCE Co., Ltd., Wakayama 640-8390, Japan; sasano@aisti.co.jp; 5Department of Internal Medicine, Kanazawa Medical University Hospital, 1-1 Daigaku, Uchinada-machi 920-0293, Japan; yagikuni@icloud.com

**Keywords:** MASLD, dihomo-γ-linolenic acid (DGLA), Δ5-desaturase (D5D), Δ6-desaturase (D6D), linoleic acid, *n*-6 polyunsaturated fatty acids, metabolomics, GC–MS, Japanese cohort

## Abstract

**Background/Objectives:** Metabolic dysfunction-associated steatotic liver disease (MASLD) has been linked to dietary fat quality and polyunsaturated fatty-acid metabolism. We evaluated whether dietary *n*-6 fatty-acid intake, serum dihomo-γ-linolenic acid (DGLA), and desaturase-based indices for Δ5-desaturase (D5D) and Δ6-desaturase (D6D) are associated with MASLD. **Methods:** We conducted a cross-sectional analysis within the HOZUGAWA health checkup cohort in Japan (n = 289; 100 MASLD, 189 non-MASLD). Participants underwent hepatic ultrasonography, dietary assessment using the Brief Self-Administered Diet History Questionnaire, and fasting serum metabolomics by gas chromatography–mass spectrometry with solid-phase dehydration derivatization. Enzyme indices were defined as the D5D index = arachidonic acid/DGLA and the D6D proxy index = DGLA/linoleic acid (hereafter referred to as the D6D index) because γ-linolenic acid was not measured. Natural-log-transformed D5D index, D6D index, DGLA, and total dietary *n*-6 fatty-acid intake were entered into multivariable logistic regression models for MASLD adjusted for age, sex, BMI, alcohol intake, and total energy. **Results:** Compared with non-MASLD, MASLD showed higher serum DGLA, lower D5D index, and higher D6D index (all *p* ≤ 0.005), with no between-group differences in total energy intake, linoleic acid, total polyunsaturated fatty acids, or total dietary *n*-6 fatty-acid intake. Higher ln D5D was independently associated with lower odds of MASLD (OR 0.62, 95% CI 0.42–0.86), whereas higher ln D6D index (OR 1.42, 95% CI 1.04–1.95) and ln DGLA (OR 1.62, 95% CI 1.13–2.43) were each positively associated. Total dietary *n*-6 fatty-acid intake was not independently associated with MASLD. **Conclusions:** In this Japanese health examination cohort, an imbalance in estimated desaturase activities—lower D5D index and higher D6D index—together with higher serum DGLA was independently associated with MASLD, whereas *n*-6 intake showed no group difference or independent association. These findings suggest that enzyme-linked endogenous *n*-6 metabolic status may be more closely related to the MASLD phenotype than intake quantity alone.

## 1. Introduction

Metabolic dysfunction-associated steatotic liver disease (MASLD) was introduced to replace and refine NAFLD nomenclature by explicitly incorporating cardiometabolic risk into the definition in addition to hepatic steatosis [1]. Beyond the quantity of fat, the quality of dietary lipids appears central to MASLD pathophysiology: randomized over-feeding trials show that excess saturated fatty acids (SFAs) preferentially increase hepatic and visceral fat, whereas excess polyunsaturated fatty acids (PUFAs) suppress hepatic fat accumulation and differentially alter body composition [2]. In youth, dietary interventions lowering the *n*-6:*n*-3 ratio have improved hepatic fat and metabolic phenotypes independently of weight loss, suggesting that qualitative fatty-acid modification may be meaningful across age groups [3]. Observational evidence further links higher circulating linoleic acid (LA), the core *n*-6 PUFA, with lower hepatic fat and reduced prevalence/incidence of fatty liver [4], while plasma profiling in MASLD has associated lower *n*-6 PUFA—particularly LA—with disease presence [5]. Collectively, these data challenge the simplistic view that “*n*-6 uniformly drives inflammation”, underscoring context, dose, and source as key determinants of effect [6].

Endogenous *n*-6 PUFA metabolism proceeds via Δ6-desaturase (D6D; LA → γ-linolenic acid [GLA]) and Δ5-desaturase (D5D; dihomo-γ-linolenic acid [DGLA] → arachidonic acid [AA]). Ratio-based indices derived from fatty-acid profiles approximate these activities (D5D index = AA/DGLA; D6D index typically = GLA/LA) and reflect an individual’s PUFA metabolic state. Across age groups, higher D5D activity is associated with favorable metabolic profiles and less hepatic fat, whereas higher D6D activity and elevated DGLA track adverse phenotypes [4,7]. Although DGLA (20:3*n*-6) can yield anti-inflammatory eicosanoids, it also sits upstream of AA, embodying dual roles in lipid biology; clinically, DGLA correlates with obesity, insulin resistance, and steatosis and has been proposed as a non-invasive biomarker for pediatric MASLD [8]. Reviews emphasize context-dependence—LA may be protective, whereas increases in GLA/DGLA or their oxylipins can be unfavorable depending on milieu [6,9]. Mechanistic work indicates that perturbing elongation/desaturation (e.g., ELOVL5 deletion) depletes long-chain PUFA, derepresses SREBP-1c, and induces fatty liver [10]; in humans, the unsaturated fatty-acid composition in liver tissue correlates with fibrosis stage, highlighting qualitative lipid features in disease progression [11].

Despite these advances, integrated, pathway-oriented analyses that concurrently evaluate (i) dietary *n*-6 intake, (ii) estimated desaturase activities, and (iii) circulating DGLA within the same individuals remain scarce—particularly in Asian health check cohorts operating under the contemporary MASLD framework. Many studies examine a single component (diet or circulating fatty acids) rather than their interplay across the pathway. Addressing this gap may clarify whether enzyme-linked metabolic imbalance is more tightly coupled to MASLD status than upstream dietary markers.

In the HOZUGAWA health check cohort, we tested whether estimated desaturase activities (Δ5-desaturase [D5D] and Δ6-desaturase [D6D]) and serum DGLA are independently associated with MASLD, while characterizing dietary patterns (including *n*-6 and *n*-3) using the BDHQ. We pre-specified enzyme indices and DGLA as pathway-level markers and evaluated their associations with MASLD after adjustment for key covariates. Our approach leverages simultaneous GC/MS with solid-phase dehydration derivatization to quantify serum metabolites on a unified platform in a real-world health check setting [12,13,14,15] (Figure 1).

## 2. Materials and Methods

### 2.1. Study Design and Ethics

We conducted a cross-sectional analysis nested within the HOZUGAWA health checkup cohort at Kameoka Municipal Hospital (Kyoto, Japan). Consecutive attendees underwent standardized procedures, including overnight fasting venipuncture, anthropometric measurements, and abdominal ultrasonography. The protocol was approved by the Ethics Committee of Kyoto Prefectural University of Medicine (approval No. ERB-C-1503) and adhered to the Declaration of Helsinki as revised in 2013. Written informed consent was obtained from all participants.

### 2.2. Participants

The enrollment window was 7 July 2020 through to 27 February 2024. Eligibility criteria were as follows: (i) availability of an overnight fasting serum sample processed for metabolomics; (ii) availability of hepatic imaging and cardiometabolic indices sufficient for MASLD classification; and (iii) availability of core covariates (age, sex, BMI, smoking status, alcohol use, and total energy intake). Participants with missing fasting serum samples or missing MASLD status were excluded by complete-case analysis (listwise deletion).

### 2.3. Outcome Measures and Assessments

#### 2.3.1. Dietary Assessment

Habitual diet during the prior month was assessed using the Brief Self-Administered Diet History Questionnaire (BDHQ) [16]. Total energy intake (kcal/day) was included as a covariate.

##### Serum Metabolomics (GC/MS with Solid Phase Dehydration Derivatization; SPDhD)

Fasting serum organic acids, amino acids, and fatty acids were profiled simultaneously by gas chromatography–mass spectrometry (GC/MS; Agilent Technologies, Santa Clara, CA, USA) after solid-phase dehydration derivatization using an inline platform (SPL M100; AiSTI SCIENCE, Wakayama, Japan) [12,13,14,15]. Identification used MS-DIAL (version 4.9.221218) against an in-house library of retention indices and EI mass spectra; quantitation used internal standard-normalized relative peak-area ratios.

Briefly, 50 µL aliquots of extract were loaded onto a stacked ion-exchange Flash-SPE ACXs cartridge, washed (acetonitrile:water = 1:1), dehydrated with acetonitrile, and nitrogen-dried on-cartridge. For organic acids and amino acids, on-cartridge methoximation (methoxyamine·HCl/pyridine) was followed by trimethylsilylation (MSTFA/hexane); derivatives were eluted in hexane and injected via a programmable-temperature large-volume injector (LVI-S250; AiSTI SCIENCE) onto a VF-5ms capillary column (30 m × 0.25 mm i.d., 0.25 µm film; Agilent Technologies, Santa Clara, CA, USA). Data were acquired in scan mode (*m*/*z* 70–600). For fatty acids, on-cartridge tert-butyldimethylsilylation (MTBSTFA/hexane) was used; GC conditions were optimized for fatty-acid derivatives (scan *m*/*z* 70–700). Analyte identification used MS-DIAL v4.9 against the in-house library; quantitation was relative to internal standards (e.g., L-norleucine for organic acids/amino acids; adipic acid for organic acids/fatty acids) and reported as analyte/IS peak-area ratios [12,13,14,15]. Fatty-acid-based enzyme indices were defined a priori as follows: D5D index = arachidonic acid (AA, 20:4*n*-6) / dihomo-γ-linolenic acid (DGLA, 20:3*n*-6) and D6D index = DGLA / linoleic acid (LA, 18:2*n*-6), because GLA (18:3*n*-6) was not measured in this platform. For analysis, indices and fatty-acid abundances were natural-log transformed as appropriate; zero values, when present, were offset by a small constant prior to log transformation.

#### 2.3.2. Data Collection

Comprehensive data were obtained through standardized questionnaires, venous blood testing, and abdominal ultrasonography as part of the routine health checkup program.

#### 2.3.3. Derived Indices and Definitions

MASLD was defined per updated international consensus guidance (2023 Delphi consensus), requiring ultrasonographic hepatic steatosis plus ≥1 cardiometabolic risk (CMR) factor [1,17,18,19]. Hepatic steatosis was assessed by abdominal ultrasonography as part of the routine health checkup program and diagnosed by trained personnel based on typical ultrasonographic features (e.g., increased liver echogenicity with hepatorenal contrast and posterior beam attenuation), following an institutional standardized protocol. CMR factors were as follows: BMI ≥ 23 kg/m^2^; fasting plasma glucose ≥ 100 mg/dL or HbA1c ≥ 5.7%; blood pressure ≥ 130/85 mm Hg; triglycerides ≥ 150 mg/dL; or HDL cholesterol ≤ 40 mg/dL (men) or ≤50 mg/dL (women). Non-excessive alcohol intake (per NAFLD/MASLD criteria) was defined as <210 g/week for men and <140 g/week for women, consistent with Japanese NAFLD/NASH clinical practice guidelines [1,17,18,19]. Because this was a community-based health checkup program, transient elastography (FibroScan) and liver biopsy were not performed; therefore, histological activity scores (e.g., the NAFLD Activity Score) and fibrosis stage could not be directly assessed. We also explored non-invasive fibrosis scores (FIB-4 and NFS). However, because these scores were largely within low-risk ranges with limited variability in this community-based cohort—and because NFS includes variables overlapping with the MASLD definition—we did not include them in the main analyses to avoid overinterpretation. Desaturase-related indices were calculated from serum fatty acid composition as follows: the Δ5-desaturase (D5D) index was defined as arachidonic acid/dihomo-γ-linolenic acid (AA/DGLA), and the Δ6-desaturase (D6D) proxy index was defined as DGLA/linoleic acid (DGLA/LA) because γ-linolenic acid (GLA) was not measured. Hereafter, the D6D proxy index is referred to as the D6D index for simplicity throughout the manuscript.

### 2.4. Statistical Analysis

All analyses were two-sided with *p* < 0.05 considered statistically significant unless otherwise noted. Analyses were performed using JMP^®^ Student Edition, Version 18.2.2 (SAS Institute Inc., Cary, NC, USA). Missing data were handled by listwise deletion. No multiplicity adjustment was applied.

#### 2.4.1. Unadjusted Between-Group Comparisons

Categorical variables were compared between MASLD and non-MASLD using Pearson’s χ^2^ test; Fisher’s exact test was used for 2 × 2 tables when any expected cell count was <5. Continuous variables were summarized as median [interquartile range] for presentation. Between-group comparisons were performed using Welch’s *t* test for approximately normally distributed variables and the Wilcoxon rank-sum test for non-normally distributed variables (two-sided α = 0.05). Reporting conventions were standardized as follows: *p* values are reported to three decimals with values <0.001 shown as “<0.001”; raw small-magnitude fatty-acid variables (DGLA, AA, LA) and the D6D index (DGLA/LA) were displayed to four decimals, and the D5D index (AA/DGLA) to two decimals; other continuous variables followed the rounding shown in each table footnote.

#### 2.4.2. Adjusted Regression Models (Primary Analyses)

We fit multivariable logistic regression models to estimate the association of desaturase indices and DGLA with MASLD. Variables defined and modeled on the log scale (e.g., lnDGLA, ln D5D index, ln D6D index) were analyzed on the log scale. For log transforms, a small offset equal to one-half of the minimum positive value within the variable was added to zeros prior to transformation. The primary exposures were ln D5D index (AA/DGLA) and ln D6D index (DGLA/LA; GLA not measured). Models were specified as follows: Model A, ln D5D index only; Model B, ln D6D index only; Model C, ln D5D index and ln D6D index entered simultaneously; and Model D, ln DGLA examined in a separate model. In addition, to evaluate whether dietary *n*-6 intake was independently associated with MASLD, we fit an additional multivariable logistic regression model including total dietary *n*-6 fatty-acid intake as the exposure. Odds ratios (ORs) were reported per 1-unit increase in the ln-transformed exposure, entered without z-standardization, with 95% CIs based on model-based (Wald) standard errors. All models were adjusted for age (years), sex (male), BMI (kg/m^2^), alcohol consumption (dummy coded with None as reference and Occasional/Daily as indicator levels), and total energy intake (kcal/day).

## 3. Results

### 3.1. Baseline and Lifestyle Characteristics

Among 289 participants (100 with MASLD and 189 without), the MASLD group included a higher proportion of men (70.0% vs. 51.3%; *p* = 0.002). Patterns of alcohol consumption (none/occasional/daily) and smoking status (never/former/current) were broadly similar between groups (Table 1), although former smoking was more frequent among participants with MASLD (27.1% vs. 17.1%). By definition, ultrasonographic hepatic steatosis was present in all individuals with MASLD but was rare among those without MASLD (Table 1).

### 3.2. Anthropometrics and Biochemical Indices

Compared with non-MASLD, participants with MASLD had higher adiposity measures, including BMI (24.9 [22.7–27.2] vs. 21.6 [20.0–23.7] kg/m^2^; *p* < 0.001) and waist circumference (Table 1). Blood pressure and glycemic indices were also less favorable in MASLD, with higher fasting glucose (107.0 [99.0–121.3] vs. 101.0 [95.0–107.0] mg/dL; *p* = 0.001) and HbA1c (6.0 [5.7–6.3] vs. 5.7 [5.6–6.0]%; *p* < 0.001). Lipid profiles showed higher triglycerides (121.5 [90.3–165.8] vs. 83.0 [61.5–112.0] mg/dL; *p* < 0.001) and lower HDL-C (58.5 [48.0–68.0] vs. 71.0 [60.0–82.5] mg/dL; *p* < 0.001), whereas total and LDL cholesterol did not differ significantly between groups. Liver enzymes (AST, ALT, and GGT) were also higher in participants with MASLD (Table 1).

### 3.3. Serum Fatty Acids and Desaturase Indices

Median serum DGLA was higher in MASLD than in non-MASLD (0.0248 [0.0148–0.0485] vs. 0.0191 [0.0122–0.0283]; *p* = 0.002). The D5D index (AA/DGLA) was lower in MASLD (4.23 [3.23–8.04] vs. 7.07 [4.60–10.61]; *p* < 0.001). Because GLA was not measured, Δ6 desaturation was proxied by the D6D index (DGLA/LA), which was higher in MASLD than in non-MASLD (0.0197 [0.0127–0.0489] vs. 0.0156 [0.0099–0.0328]; *p* = 0.005) (Table 1).

Values are n (%) for categorical variables and median [IQR] for continuous variables, unless otherwise indicated. Abbreviations: ALT, alanine aminotransferase; AST, aspartate aminotransferase; GGT, γ-glutamyl transferase; TBDMS, tert-butyldimethylsilyl (derivative). Categorical variables (sex, alcohol consumption [None/Occasional/Daily], smoking status [Never/Former/Current]) were compared between MASLD and non-MASLD groups using Pearson’s χ^2^ test; when any expected cell count was <5 in 2 × 2 tables, Fisher’s exact tests were used (two-sided α = 0.05). Continuous variables, including clinical laboratory indices and fatty-acid metabolites, were compared using Welch’s *t* test (two-sided) or the Wilcoxon rank-sum test, as appropriate. Fatty-acid indices were defined as follows: D5D index = AA/DGLA and D6D index = DGLA/LA, because γ-linolenic acid (18:3*n*-6) was not measured. Missing data were handled by listwise deletion, and no multiplicity adjustment was applied.

### 3.4. Dietary Intake (BDHQ)

Total energy intake and macro- and micronutrient intakes—including protein, fat, carbohydrates, total/saturated/mono/polyunsaturated fatty acids, and dietary *n*-6 and *n*-3 fatty acids—did not differ between groups (all *p* > 0.05; Table 2).

Values are median [IQR] estimated from the BDHQ. Continuous variables were compared between groups using Welch’s *t* test or the Wilcoxon rank-sum test, as appropriate (two-sided α = 0.05). Missing data were handled by listwise deletion; no multiplicity adjustment was applied.

### 3.5. Adjusted Associations

In multivariable logistic models adjusted for age, sex, BMI, alcohol intake, and total energy, higher ln D5D index was associated with lower odds of MASLD (Model A: OR, 0.62; 95% CI, 0.42–0.86), whereas higher ln D6D index was associated with higher odds (Model B: OR, 1.42; 95% CI, 1.04–1.95). When both indices were included simultaneously (Model C), associations remained directionally consistent (ln D5D index, OR 0.61 [95% CI 0.42–0.84]; ln D6D index, OR 1.52 [95% CI 1.08–2.13]). In a separate model, ln DGLA was positively associated with MASLD (Model D: OR, 1.62; 95% CI, 1.13–2.43) (Table 3). By contrast, when total dietary *n*-6 fatty acid intake was entered as the exposure and adjusted for age, sex, BMI, alcohol intake, and total energy intake, no significant association with MASLD was observed (*p* = 0.929).

Model A includes ln D5D index (AA/DGLA) as the sole exposure; Model B includes ln D6D index (DGLA/LA; GLA not measured); Model C includes ln D5D index and ln D6D index simultaneously; Model D includes ln DGLA in a separate model. Odds ratios (ORs) and 95% CIs are reported per 1-unit increase in the ln-transformed exposure (natural-log transformed and entered without z-standardization). All models are adjusted for age (years), sex (male), BMI (kg/m^2^), alcohol consumption (Occasional/Daily vs. None), and total energy intake (kcal/day), consistent with the prespecified analysis plan (Methods 2.4). Two-sided tests were used with *p* < 0.05 considered statistically significant; *p* < 0.001 is shown as “<0.001”. Missing data were handled by listwise deletion; no multiplicity adjustment was applied. MASLD was defined per the 2023 consensus (ultrasonographic steatosis plus ≥1 cardiometabolic risk factor); the enzyme indices were the D5D index = AA/DGLA and D6D index = DGLA/LA. See Results for numerical estimates.

Abbreviations: AA, arachidonic acid; BMI, body mass index; CI, confidence interval; D5D, Δ5-desaturase; D6D, Δ6-desaturase; DGLA, dihomo-γ-linolenic acid; GLA, γ-linolenic acid; LA, linoleic acid; MASLD, metabolic dysfunction-associated steatotic liver disease; OR, odds ratio; SD, standard deviation.

## 4. Discussion

This study is unique in that it simultaneously evaluated serum DGLA (20:3*n*-6), estimated desaturase activity (D5D index = AA/DGLA, D6D index = DGLA/LA), and habitual diet within the same analytical framework in a Japanese health examination cohort. Three observations emerged. First, at the group level, MASLD was characterized by higher DGLA, lower ln D5D index, and higher ln D6D index. Second, in multivariable models—adjusted for age, sex, BMI, alcohol intake, and total energy—higher ln D5D was associated with lower odds of MASLD (e.g., OR 0.61–0.62), whereas higher ln D6D index was associated with higher odds (OR 1.42–1.52). Third, ln DGLA showed a positive independent association with MASLD when modeled separately (OR 1.62). Collectively, the lack of association between total dietary *n*-6 fatty acid intake and MASLD after multivariable adjustment, together with the robust associations of D5D index, D6D index, and serum DGLA with MASLD, indicates that imbalances in endogenous *n*-6 metabolism (desaturase activities and DGLA accumulation), rather than differences in total *n*-6 intake per se, are more closely linked to MASLD in this setting.

The pattern observed here aligns with evidence that fat quality—not only quantity—modulates hepatic lipid handling: overfeeding trials show SFA preferentially raise hepatic/visceral fat, whereas PUFA suppress hepatic fat [2], and pediatric interventions lowering the *n*-6:*n*-3 ratio improve the hepatic and metabolic phenotypes [3]. Epidemiology consistently links higher circulating linoleic acid (LA) with lower fatty-liver risk [4] and shows relative LA depletion in MASLD [5], tempering the notion that “*n*-6 is uniformly harmful” and emphasizing context-dependence [6]. Within this context, our findings that desaturase imbalance tracks MASLD—lower D5D and higher D6D with elevated DGLA—are congruent with reports across age groups in which higher D5D relates to favorable metabolic profiles and higher D6D/DGLA to adverse phenotypes [4,7]. While DGLA can yield anti-inflammatory eicosanoids, its elevation may alternatively reflect a Δ5-desaturation bottleneck, consistent with its correlations with obesity, insulin resistance, and steatosis, and its proposed utility as a pediatric biomarker [8].

In the *n*-6 pathway (LA → GLA → DGLA via D6D; DGLA → AA via D5D), the combination of elevated DGLA + reduced D5D + elevated D6D suggests a reduced conversion efficiency to AA and retention of the intermediate. Perturbations that limit long-chain PUFA supply can derepress SREBP-1c, favor de novo lipogenesis, and promote steatosis; for example, ELOVL5 deletion induces fatty liver in mice [10]. In humans, hepatic unsaturated fatty-acid composition relates inversely to fibrosis severity [11]. These data support a working model in which qualitative lipid features and desaturase balance influence MASLD biology via membrane composition, oxylipin profiles, and lipid-peroxidation dynamics.

Desaturase-based markers may have value for risk stratification or phenotyping: a higher ln D5D index corresponded to lower odds of MASLD, whereas a higher ln D6D index corresponded to higher odds, and ln DGLA—interpreted carefully given its biochemical coupling to D5D—also showed a positive association when assessed independently. Notably, we did not co-enter the ln DGLA with ln D5D index to avoid collinearity induced by the shared numerator/denominator structure, evaluating DGLA in a separate model by design. Our null between-group differences in dietary *n*-6 (BDHQ) do not argue against nutritional guidance; rather, they reinforce that “quality over quantity” remains relevant—reducing SFA and balancing with *n*-3 is a plausible target [2,3]—while enzyme-linked metabolic status may be more proximate to the MASLD phenotype than total *n*-6 intake alone [4,6].

### 4.1. Limitations

This study has several limitations. First, because this was a community-based health checkup cohort, transient elastography and liver biopsy were not performed; therefore, histological disease activity (e.g., the NAFLD Activity Score) and fibrosis stage could not be directly assessed. Future studies incorporating elastography and/or histology are needed to evaluate disease severity and fibrosis in relation to biochemical markers. Although non-invasive fibrosis scores (the FIB-4 index and the NAFLD fibrosis score [NFS]) were available, they were largely within low-risk ranges with limited variability in this community-based health checkup cohort; moreover, NFS includes components overlapping with the MASLD definition (e.g., BMI and glycemic status). Therefore, we did not include these scores in the main analyses to avoid overinterpretation and redundancy. Second, the cross-sectional design precludes causal inference, leaving potential for reverse causality and residual confounding. Third, relative quantification requires caution regarding the external transferability of absolute amounts to other cohorts. Fourth, GLA (18:3*n*-6) was not measured in this study; thus, D6D was proxied by DGLA/LA. Additional limitations include measurement error in dietary assessment (BDHQ), constraints on generalizability due to a single-center health examination cohort, and unadjusted multiple testing.

### 4.2. Perspective

Future work should incorporate absolute quantification and fraction-specific lipidomics, capture true D6D (GLA), and test longitudinal change to clarify temporal ordering. Genetic (e.g., FADS1/2) or intervention approaches may further delineate whether desaturase imbalance is a driver or marker of MASLD biology [4,7,10,11].

## 5. Conclusions

In this Japanese health examination cohort, an imbalance in estimated desaturase activities—lower D5D and higher D6D were independently associated with MASLD, and serum ln DGLA—also showed a positive association when evaluated in a separate model. By contrast, dietary *n*-6 intake did not differ between groups, supporting the view that enzyme-linked metabolic status may be closer to the MASLD phenotype than intake quantity alone. Future work should enhance the clinical applicability of desaturase-based markers (e.g., the D5D/D6D balance) by incorporating absolute quantification, direct estimation of D6D (with GLA measured), and longitudinal validation across diverse populations.

## Figures and Tables

**Figure 1 nutrients-18-00057-f001:**
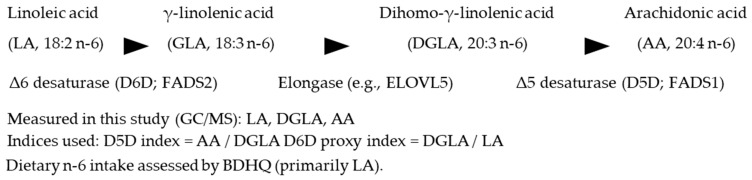
Schematic overview of the *n*-6 polyunsaturated fatty-acid (PUFA) metabolic pathway and definitions of desaturase-related indices used in this study. Linoleic acid (LA) is converted to γ-linolenic acid (GLA) by Δ6-desaturase (D6D), elongated to dihomo-γ-linolenic acid (DGLA), and further converted to arachidonic acid (AA) by Δ5-desaturase (D5D). The D5D index was defined as AA/DGLA. Because GLA was not measured in this study, the D6D proxy index was defined as DGLA/LA. Abbreviations: PUFA, polyunsaturated fatty acid; LA, linoleic acid; GLA, γ-linolenic acid; DGLA, dihomo-γ-linolenic acid; AA, arachidonic acid; D5D, Δ5-desaturase; D6D, Δ6-desaturase; FADS1, fatty acid desaturase 1; FADS2, fatty acid desaturase 2; ELOVL5, elongation of very long-chain fatty acids protein 5; BDHQ, Brief-type Self-Administered Diet History Questionnaire.

**Table 1 nutrients-18-00057-t001:** Participant characteristics, clinical laboratory indices, and fatty-acid metabolites by MASLD status.

Characteristics	n (Non-MASLD)		Non-MASLD, Median [IQR]	n (MASLD)	MASLD, Median [IQR]	*p* Value
Men	97/189 (51.3%)			70/100 (70.0%)		0.002
Age, years	189		68 [61, 73]	100	68 [59.3, 71.8]	0.485
Body mass index, kg/m^2^	189		21.6 [20.0, 23.7]	100	24.9 [22.7, 27.2]	<0.001
Waist circumference, cm	189		82.3 [76.3, 87.3]	100	91.3 [85.0, 96.5]	<0.001
Systolic blood pressure, mmHg	189		132.5 [122.0, 145.0]	100	138.0 [126.5, 149.3]	0.018
Diastolic blood pressure, mmHg	189		79.0 [71.8, 89.0]	100	83.0 [75.3, 92.0]	0.009
Glucose, mg/dL	189		101.0 [95.0, 107.0]	100	107.0 [99.0, 121.3]	0.001
Hemoglobin A1c, %	189		5.7 [5.6, 6.0]	100	6.0 [5.7, 6.3]	<0.001
Total cholesterol, mg/dL	189		208.0 [187.0, 233.0]	100	201.0 [181.3, 224.0]	0.056
Triglycerides, mg/dL	189		83.0 [61.5, 112.0]	100	121.5 [90.3, 165.8]	<0.001
HDL cholesterol (HDL-C), mg/dL	189		71.0 [60.0, 82.5]	100	58.5 [48.0, 68.0]	<0.001
LDL cholesterol (LDL-C), mg/dL	189		116.2 [99.2, 136.4]	100	112.8 [88.8, 138.1]	0.459
AST, U/L	189		21.0 [18.0, 25.0]	100	23.0 [18.0, 30.5]	0.022
ALT, U/L	189		17.0 [14.0, 20.0]	100	24.0 [18.0, 31.0]	<0.001
GGT, U/L	189		20.0 [14.0, 27.0]	100	31.0 [22.0, 52.0]	<0.001
Dihomo-gamma-linolenic acid(C20:3*n*-6)_TBDMS	186		0.0191 [0.0122, 0.0283]	98	0.0248 [0.0148, 0.0485]	0.002
Arachidonic acid(C20:4*n*-6)_TBDMS	186		0.1547 [0.0971, 0.2088]	98	0.1392 [0.0909, 0.1995]	0.230
Linoleic acid(C18:2*n*-6)_TBDMS	186		1.2804 [0.8638, 1.5887]	98	1.2624 [0.9099, 1.5196]	0.749
D5D index	186		7.07 [4.60, 10.61]	98	4.23 [3.23, 8.04]	<0.001
D6D index	186		0.0156 [0.0099, 0.0328]	98	0.0197 [0.0127, 0.0489]	0.005
Alcohol consumption						
None	80/175 (45.7%)			43/95 (45.3%)		0.663
Occasional	42/175 (24.0%)			19/95 (20.0%)		
Daily	53/175 (30.1%)			33/95 (34.7%)		
Smoking status						
Never	123/175 (70.3%)			59/96 (61.5%)		0.154
Former	30/175 (17.1%)			26/96 (27.1%)		
Current	22/175 (12.6%)			11/96 (11.4%)		
Hepatic steatosis (Yes)	2/189 (1.1%)			100/100 (100.0%)		<0.001

**Table 2 nutrients-18-00057-t002:** Energy and macronutrient intakes (BDHQ) by MASLD status.

Dietary Intake	Non-MASLD	MASLD	*p* Value
*n*	178	93	
Energy intake, kcal/day	1715.9 [1363.0, 2091.5]	1923.1 [1553.0, 2257.2]	0.111
Protein intake, g/day	67.9 [53.8, 87.7]	69.4 [55.3, 86.6]	0.760
Animal protein intake, g/day	40.4 [30.1, 53.3]	38.5 [28.2, 50.5]	0.512
Plant protein intake, g/day	27.8 [22.9, 35.2]	29.1 [23.8, 35.0]	0.630
Lipids intake, g/day	53.6 [42.8, 65.9]	55.4 [40.8, 69.0]	0.715
Animal lipids intake, g/day	24.1 [17.7, 33.1]	25.6 [18.7, 31.3]	0.626
Plant lipids intake, g/day	27.4 [21.8, 33.9]	28.3 [21.4, 36.1]	0.866
Carbohydrates intake, g/day	213.9 [170.0, 267.8]	234.8 [199.1, 305.7]	0.082
Saturated fatty acids, g/day	14.4 [10.7, 17.9]	14.3 [10.9, 17.5]	0.884
Monounsaturated fatty acids, g/day	18.9 [14.7, 23.2]	19.3 [14.6, 24.5]	0.970
Polyunsaturated fatty acids, g/day	12.6 [10.2, 16.0]	13.4 [9.8, 16.8]	0.645
Cholesterol, mg/day	403.9 [282.3, 525.8]	392.2 [267.6, 511.3]	0.543
Water-soluble dietary fiber, g/day	3.0 [2.1, 3.9]	2.8 [2.2, 3.6]	0.550
Insoluble dietary fiber, g/day	8.3 [5.8, 10.6]	8.2 [6.5, 10.2]	0.814
Total dietary fiber, g/day	11.6 [8.0, 14.8]	11.6 [8.7, 14.4]	0.728
Salt equivalent, g/day	10.2 [8.2, 13.0]	11.0 [8.6, 13.9]	0.195
Sucrose, g/day	12.2 [8.2, 18.2]	11.3 [7.1, 16.2]	0.085
Alcohol, g/day	0.9 [0, 16.7]	1.2 [0, 31.1]	0.280
*n*-3 fatty acids, g/day	2.6 [1.9, 3.5]	2.7 [2.0, 3.6]	0.859
*n*-6 fatty acids, g/day	10.0 [8.1, 12.1]	10.5 [7.9, 13.4]	0.594

**Table 3 nutrients-18-00057-t003:** Adjusted associations of desaturase indices and DGLA with MASLD in multivariable logistic models.

Model	Predictor	n	OR (95% CI)
Model A: ln D5D index	ln D5D index	249	0.62 (0.42–0.86)
Model B: ln D6D index	ln D6D index	249	1.42 (1.04–1.95)
Model C: ln D5D index + ln D6D index	ln D5D index	249	0.61 (0.42–0.84)
Model C: ln D5D index + ln D6D index	ln D6D index	249	1.52 (1.08–2.13)
Model D: ln DGLA	ln DGLA	249	1.62 (1.13–2.43)

## Data Availability

The data presented in this study are available on request from the corresponding author due to participant privacy and institutional policies.

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
