# Peer review of "Dihomo-γ-Linolenic Acid Elevation with Desaturase Imbalance in Metabolic Dysfunction-Associated Steatotic Liver Disease in a Japanese Health Checkups Cohort: HOZUGAWA Study, a Multi-Omic, Diet Adjusted Analysis"

_nutrients, 2025, doi:10.3390/nu18010057_

Round 1
Reviewer 1 Report
Comments and Suggestions for Authors
Review for the manuscript ,, Dihomo γ linolenic Acid Elevation with Desaturase Imbalance in Metabolic dysfunction-associated steatotic liver disease in a Japanese Health checkups Cohort: HOZUGAWA study, a Multi-omic, Diet Adjusted Analysis,,
Metabolic dysfunction-associated steatotic liver disease (MAFLD) is the most prevalent liver disease globally, with an ever-increasing incidence.The increasing global prevalence of MS amplifies the cirrhotic risk of fatty liver disease, transforming it into a significant public health problem. Continuing research in this area is essential to better understand pathogenic mechanisms and develop effective treatments.
In this manuscript the authors determined the role of dietary n-6 fatty-acid intake, serum dihomo-γ-linolenic acid (DGLA), and desaturase-based indices for Δ5-desaturase (D5D) and Δ6-desaturase (D6D) for the development of MASLD.
Comments
What type of methods did the authors used to diagnoses MASLD? Did they use fibro-scan elastography or hepatic biopsy?
I think it would be interesting to evaluate the NAFLD Activity Score (steatosis, lobular inflammation, hepatocyte ballooning) to determine the severity of liver damage and to track the progression of the disease.
Also, it should be interesting if the authors evaluate the degree of liver fibrosis in association with biochemical determinations.
Author Response
Dear Reviewers,
We sincerely thank you for your constructive comments on our manuscript (Manuscript ID: nutrients-4047604). We have revised the manuscript accordingly. We provide a point-by-point response below.
Response to Reviewer 1
Comments
What type of methods did the authors used to diagnoses MASLD? Did they use fibro-scan elastography or hepatic biopsy?
I think it would be interesting to evaluate the NAFLD Activity Score (steatosis, lobular inflammation, hepatocyte ballooning) to determine the severity of liver damage and to track the progression of the disease.
Also, it should be interesting if the authors evaluate the degree of liver fibrosis in association with biochemical determinations.
General Response
We thank the reviewer for the constructive and insightful comments. We have revised the manuscript accordingly, clarified the diagnostic approach for MASLD in our health-checkup cohort, and added a dedicated limitations statement regarding disease severity assessment. We also considered non-invasive fibrosis scores (FIB-4 and NFS); however, in this community-based health-checkup cohort these scores were largely within low-risk ranges with limited variability, and NFS includes components overlapping with the MASLD definition (e.g., BMI and glycemic status). Therefore, we did not include these scores in the main analyses to avoid redundancy and overinterpretation; instead, we explicitly addressed this in the Discussion (Limitations) and highlighted the need for future studies incorporating elastography and/or histology.
Response
Thank you for this important question. In this health-checkup cohort, MASLD was diagnosed based on ultrasonographic evidence of hepatic steatosis plus ≥1 cardiometabolic risk factor, consistent with the 2023 Delphi consensus. We have expanded the Methods to describe in detail how hepatic steatosis was assessed by abdominal ultrasonography (trained personnel, standardized protocol, and diagnostic ultrasonographic features). Because this was a community-based screening program, transient elastography and liver biopsy were not performed; therefore, histological assessments, including the NAFLD Activity Score (NAS), and direct evaluation of fibrosis stage were not possible. We additionally note that although FIB-4 and NFS were available, they were not included in the main analyses for the reasons described above and are discussed as a limitation. We have now explicitly clarified these points and acknowledged them as limitations in the revised manuscript.
Changes made
- We added a detailed description of (i) how hepatic steatosis was assessed by abdominal ultrasonography (trained personnel, standardized protocol, and diagnostic features), (ii) cardiometabolic risk factors used to define MASLD, and (iii) the definition of non-excessive alcohol intake (sex-specific thresholds) used for MASLD classification, as well as the handling of missing data (complete-case analysis).
- We clarified that transient elastography (FibroScan) and liver biopsy were not performed; therefore, histological activity scores (including NAS) and direct assessment of fibrosis stage were not possible. These points are now explicitly stated and acknowledged as limitations in the revised manuscript.
Location:
Materials and Methods, Line [144]–[147], [151]-[158]. Discussion, Lines [315]–[323].
In addition, we corrected the labeling in Table 3 to reflect the actual modeling approach: exposures were ln-transformed and entered without z-standardization, and ORs are now described per 1-unit increase in the ln-transformed exposure (we removed “per +1 SD” and “(z)” from the table/footnote).
No changes were made to the underlying model estimates.
Location
Table 3, Line [190]-[192], [256]-[258].
We thank the reviewers for their constructive comments, which have helped us improve the clarity and quality of the manuscript. We hope that the revised version adequately addresses all concerns. We would be pleased to further revise the manuscript if additional changes are needed.

Reviewer 2 Report
Comments and Suggestions for Authors
The authors evaluated a wide range of biochemical and anthropometric measurements in participants with and without MASLD, with a particular focus on dietary n-6 fatty acid intake, serum dihomo-γ-linolenic acid levels, and desaturase-based indices related to Δ5- and Δ6-desaturase activity. The manuscript is generally well structured, and the conclusions are consistent with the stated objectives. However, several points merit consideration, as outlined below:
(1) The inclusion of a schematic figure illustrating the metabolism of unsaturated fatty acids relevant to the study could improve the clarity and ease of interpretation of the results.
(2) The subsection title is conceptually inconsistent with the statistical methods described, as Pearson’s chi-square and Fisher’s exact tests are inferential procedures rather than purely descriptive analyses.
(3) The n_MASLD column in Table 2 appears unnecessary, as all values are identical.
(4) Some portions of the text unnecessarily repeat numerical values already presented in the tables.
(5) Several biochemical indices reported in Table 1 are not adequately described or discussed in the main text.
Author Response
Dear Reviewers,
We sincerely thank you for your constructive comments on our manuscript (Manuscript ID: nutrients-4047604). We have revised the manuscript accordingly. We provide a point-by-point response below.
Response to Reviewer 2
General Response
We sincerely thank the reviewer for the helpful and detailed suggestions. We have revised the manuscript to improve clarity and presentation, including adding a schematic figure of unsaturated fatty acid metabolism, refining section titles to match the statistical methods, simplifying tables, reducing redundant numerical repetition, and expanding the explanation of biochemical indices.
(1) Comments
The inclusion of a schematic figure illustrating the metabolism of unsaturated fatty acids relevant to the study could improve the clarity and ease of interpretation of the results.
Response
Thank you for this helpful suggestion. To improve clarity and facilitate interpretation, we added a schematic figure illustrating the n-6 PUFA metabolic pathway (LA→GLA→DGLA→AA) and clearly defining the desaturase-related indices used in this study (D5D index = AA/DGLA; D6D proxy index = DGLA/LA because GLA was not measured).
Changes made
A new schematic figure has been added to the revised manuscript (Figure 1) together with an explanatory legend and abbreviation definitions. In addition, we clarified in the Abstract and in the “Derived indices and definitions” subsection that the D6D index used in this study is a proxy (DGLA/LA) due to the lack of GLA measurement, and we refer to it as the D6D index thereafter for simplicity.
Location:
Figure 1, Lines [88]-[95].
Abstract Lines [25]-[27]. Introduction Lines [62].
Materials and Methods, Lines [161]-[162].
(2) Comments
The subsection title is conceptually inconsistent with the statistical methods described, as Pearson’s chi-square and Fisher’s exact tests are inferential procedures rather than purely descriptive analyses.
Response
We agree with this comment. Pearson’s χ² test and Fisher’s exact test are inferential procedures for comparing groups. We therefore revised the subsection title to better reflect the content.
Change made:
The subsection title in the Statistical Analysis section was changed from “Descriptive comparisons” to “Unadjusted between-group comparisons.”
Location:
Materials and Methods, Lines [167].
(3) Comments
The n_MASLD column in Table 2 appears unnecessary, as all values are identical.
Response
We agree. Because the n_MASLD column was redundant (identical across rows), we removed this column. To improve clarity, we now report the sample size for each group as the first row of Table 2 (n for Non-MASLD and MASLD).
Changes made
The redundant n_MASLD column was removed from Table 2, and a first-row “n” entry was added to display the number of participants in each group.
Location
Table 2.
(4) Comments
Some portions of the text unnecessarily repeat numerical values already presented in the tables.
Response:
Thank you for this helpful suggestion. We revised the Results section to reduce redundancy by removing extensive numerical listings that duplicated the information in Tables 1–3. The revised text now focuses on the key comparisons and overall patterns, while directing readers to the tables for complete descriptive statistics and detailed estimates.
Changes made:
We streamlined the Results text throughout (Baseline and Lifestyle Characteristics; Anthropometrics and Biochemical Indices; Serum Fatty Acids and Desaturase Indices; Dietary Intake; and Adjusted Associations) and replaced repeated numerical values with concise summaries and table references.
Location: Results section, Lines [198]–[220], [235]-[237].
(5) Comments
Several biochemical indices reported in Table 1 are not adequately described or discussed in the main text.
Response
We agree with this comment. Uric acid and renal function indices (creatinine and eGFR) were not central to our primary objective focused on n-6 PUFA metabolism and desaturase-related indices; therefore, we removed them from Table 1 to streamline the presentation. In addition, for key liver-related biochemical indices retained in Table 1 (AST, ALT, and GGT), we added a brief statement in the Results to orient readers without repeating numerical values.
Changes made
We removed uric acid, creatinine, and eGFR from Table 1. We also added one sentence in the Results noting that AST/ALT/GGT were higher in MASLD (Table 1), without repeating numerical values.
Location
Table 1
In addition, we corrected the labeling in Table 3 to reflect the actual modeling approach: exposures were ln-transformed and entered without z-standardization, and ORs are now described per 1-unit increase in the ln-transformed exposure (we removed “per +1 SD” and “(z)” from the table/footnote).
No changes were made to the underlying model estimates.
Location
Table 3, Line [190]-[192], [256]-[258].
We thank the reviewers for their constructive comments, which have helped us improve the clarity and quality of the manuscript. We hope that the revised version adequately addresses all concerns. We would be pleased to further revise the manuscript if additional changes are needed.
